# Genome-Wide Identification and Expression Analysis of the Sucrose Synthase Gene Family in Sweet Potato and Its Two Diploid Relatives

**DOI:** 10.3390/ijms241512493

**Published:** 2023-08-06

**Authors:** Zhicheng Jiang, Huan Zhang, Shaopei Gao, Hong Zhai, Shaozhen He, Ning Zhao, Qingchang Liu

**Affiliations:** Key Laboratory of Sweetpotato Biology and Biotechnology, Ministry of Agriculture and Rural Affairs/Beijing Key Laboratory of Crop Genetic Improvement/Laboratory of Crop Heterosis and Utilization, Ministry of Education, College of Agronomy & Biotechnology, China Agricultural University, Beijing 100193, China; pyxjzc@163.com (Z.J.); zhanghuan1111@cau.edu.cn (H.Z.); spgao@cau.edu.cn (S.G.); zhaihong@cau.edu.cn (H.Z.); sunnynba@cau.edu.cn (S.H.); zhaoning2012@cau.edu.cn (N.Z.)

**Keywords:** sweet potato, *Ipomoea trifida*, *Ipomoea triloba*, sucrose synthase, storage root development, starch biosynthesis, abiotic stress, hormone crosstalk

## Abstract

Sucrose synthases (SUS; EC 2.4.1.13) encoded by a small multigene family are the central system of sucrose metabolism and have important implications for carbon allocation and energy conservation in nonphotosynthetic cells of plants. Though the *SUS* family genes (*SUSs*) have been identified in several plants, they have not been explored in sweet potato. In this research, nine, seven and seven *SUSs* were identified in the cultivated sweet potato (*Ipomoea batatas*, 2*n* = 6*x* = 90) as well as its two diploid wild relatives *I. trifida* (2*n* = 2*x* = 30) and *I. triloba* (2*n* = 2*x* = 30), respectively, and divided into three subgroups according to their phylogenetic relationships. Their protein physicochemical properties, chromosomal localization, phylogenetic relationship, gene structure, promoter *cis*-elements, protein interaction network and expression patterns were systematically analyzed. The results indicated that the *SUS* gene family underwent segmental and tandem duplications during its evolution. The *SUSs* were highly expressed in sink organs. The *IbSUSs* especially *IbSUS2*, *IbSUS5* and *IbSUS7* might play vital roles in storage root development and starch biosynthesis. The *SUSs* could also respond to drought and salt stress responses and take part in hormone crosstalk. This work provides new insights for further understanding the functions of *SUSs* and candidate genes for improving yield, starch content, and abiotic stress tolerance in sweet potatoes.

## 1. Introduction

Sucrose is mainly synthesized in leaves and green parts of plants and then transported to sink tissues where it serves as a carbon source of energy for various metabolic pathways [1,2,3,4,5]. Sucrose synthase (SUS; EC 2.4.1.13), which reversibly catalyzes the conversion of sucrose to fructose and UDP-glucose, is considered as the key enzyme involved in the sugar metabolism of plants [2]. As a result, sucrose biosynthesis is essential for the growth and development of plants [2,6,7]. SUS proteins (SUSs) are usually homotetramers composed of subunits with a molecular mass of about 90 kDa and belong to the glycosyltransferase-4 subfamily of glycosyltransferases [4,8,9]. SUSs are encoded by a small multigene family [10]. The SUS family genes (*SUSs*) have been identified in several plants. For example, six, three, and six *SUSs* were identified in *Arabidopsis* [11], maize [12], and rice [13], respectively. The genomes of *Gossypium arboreum*, *G. raimondii* and *G. hirsutum* contained eight, eight, and 15 *SUSs*, respectively [14]. Fifteen *SUSs* were identified in poplar [15].

There are currently several reports on the functions of *SUSs* in plants. Potato *Sus3* gene was highly expressed in stems and roots, which showed the vascular function of SUS, and the *Sus4* gene was expressed primarily in the storage and vascular tissues of tubers, which facilitated the sink function of SUS [16]. Biemelt et al. [17] suggest that during hypoxia SUS might function in channeling excess carbohydrates into cell wall polymers for later consumption in potatoes. The two maize isozymes of SUS were found to be critical in endosperm development, SS1 for cell wall integrity and SS2 for starch biosynthesis [18]. The study of Ricard et al. [19] proved the critical role of SUS for anoxic tolerance of maize roots. The endosperm-specific transcription factor Opaque2 (O2) transactivated *Sus1* and *Sus2* and regulated starch biosynthesis during endosperm filling in maize [20]. In *Arabidopsis*, SUS possibly had a role in carbon partitioning during the early to middle stages of seed development, and at later stages, it might play roles within the embryo, cotyledon, and aleurone layer [21]. The results of Baroja-Fernándeza et al. [22] indicated that SUS was involved in the cellulose and starch biosynthesis of *Arabidopsis*. *AtSUS3* in *Arabidopsis* regulated starch accumulation in guard cells [23]. Overexpression of a potato *SUS* gene in cotton elevated SUS activity, improved early seed development, increased seed set, and promoted fiber elongation [24]. In cotton, GhSUS2 positively regulated fiber development, but its phosphorylation by GhCPK84 and GhCPK93 inhibited fiber elongation [25]. Thermo-responsive *Sus3* gene was involved in high-temperature tolerance during the ripening stage in rice [26]. Overexpression of the bamboo *BeSUS5* gene in poplar improved cellulose content, cell wall thickness, and fiber quality [27].

Sweet potato (*Ipomoea batatas* (L.) Lam., 2*n* = B_1_B_1_B_2_B_2_B_2_B_2_ = 6*x* = 90) is an autohexaploid crop that belongs to the family Convolvulaceae, Genus *Ipomoea*, and Section *Batatas*. It is an economically important starchy root crop worldwide used as food, industrial materials, and bioenergy resource [28,29,30]. As the main component of sweet potato, starch accounts for 50–80% of its dry matter content [31]. As an important part of carbon metabolism, it is necessary to investigate the biological functions and regulatory mechanisms of *SUSs* in sweet potato. Recently, as the release of genome assemblies of cultivated sweet potato [32] and its two diploid wild relatives *Ipomoea trifida* and *I. triloba* [33], it is possible to identify and analyze the SUS gene family at the whole genome level in sweet potato. According to Zhang et al. [34], nine *SUSs* existed in the genome of sweet potato and the expression level of *IbSus6* was higher in the storage roots of Kokei 14 than in those of its mutant. However, the characteristics, functions and regulatory mechanisms of *SUSs* remain unclear in sweet potato.

In the present study, *SUSs* were identified from sweet potato (*I*. *batatas*), *I. trifida*, and *I*. *triloba*, and then their protein physicochemical properties, chromosomal localization, phylogenetic relationships, gene structure, promoter *cis*-elements and protein interaction network were systematically analyzed. Their tissue specificity and expression pattern analyses for storage root development, starch biosynthesis and abiotic stress and hormone responses were carried out using qRT-PCR or RNA-seq. The aim of this study was to provide new insights for further understanding the functions of *SUSs* and candidate genes for improving yield, starch content, and abiotic stress tolerance in sweet potatoes.

## 2. Results

### 2.1. Identification of SUSs in Sweet Potato and Its Two Diploid Wild Relatives

In order to completely identify *SUSs* in genomes of cultivated sweet potato and its two diploid wild relatives *I. trifida* and *I*. *triloba*, three typical strategies (i.e., blastp search, hmmer search, and the CD-search database) were used. Nine, seven, and seven *SUSs* were separately identified in *I. batatas*, *I. trifida* and *I. triloba*, which were named “*IbSUS*”, “*ItfSUS*”, and “*ItbSUS*”, respectively. Their basic characteristics were analyzed using the sequences from *I. batatas*, *I. trifida* and *I. triloba* (Table 1). The CDS length of *IbSUSs* ranged from 2232 bp to 2721 bp and the genomic length varied from 4091 bp to 6116 bp. The length of putative proteins ranged from 743 aa to 906 aa, with a molecular weight (MW) of 84.78 kDa to 101.58 kDa and a theoretical isoelectric point (*p*I) of 6.02 to 6.99. Most of IbSUSs were stable with an instability index of less than 40 except for IbSUS1 which obtained an instability index of 40.26. The grand average of hydropathicity (GRAVY) values of IbSUSs varied from −0.351 to −0.202, indicating that they are hydrophilic proteins. Subcellular localization prediction showed that IbSUS1, IbSUS3, IbSUS4, and IbSUS9 were located in the cytoplasm, IbSUS2 and IbSUS8 in chloroplasts, IbSUS6 and IbSUS7 in mitochondria, and IbSUS5 in plastids (Table 1).

In *I. trfida*, the CDS length ranged from 2418 bp to 2679 bp, and the genomic length varied from 4194 bp to 6346 bp. The length of putative proteins ranged from 805 aa to 892 aa, with MW of 92.21 kDa to 99.89 kDa and *p*I of 5.93 to 6.40. All ItfSUSs were stable with an instability index of less than 40. GRAVY values varied from −0.336 to −0.248, indicating that they are hydrophilic proteins. Most of the ItfSUSs were located in the cytoplasm except ItfSUS2 and ItfSUS5 located in chloroplasts (Table 1). In *I. triloba*, the CDS length ranged from 2418 bp to 2664 bp and the genomic length varied from 4028 bp to 6636 bp. The length of putative proteins ranged from 805 aa to 887 aa, with MW of 92.09 kDa to 99.29 kDa and *p*I of 5.93 to 6.51. All ItfSUSs were stable with an instability index of less than 40. GRAVY values were less than 0, which are hydrophilic. ItbSUS1, ItbSUS4, ItbSUS6, and ItbSUS7 were located in the cytoplasm and ItbSUS2, ItbSUS3 and ItbSUS5 located in chloroplasts (Table 1).

Chromosomal localization showed that all of the *SUSs* from *I. batatas*, *I. triloba*, and *I. trifida* were distributed across six chromosomes (Figure 1). In *I. batatas*, three *IbSUSs* were detected on LG13, two on LG7, one on LG1, LG2, LG8 and LG15, but no *SUSs* were detected on LG3, LG4, LG5, LG6, LG9, LG10, LG11, LG12 and LG14 (Figure 1a). In *I. trifida* and *I. triloba*, similar distributions of *SUSs* were found on Chr02 (2), Chr03 (1), Chr04 (1), Chr05(1), Chr06 (1) and Chr11 (1) (Figure 1b,c). Furthermore, we analyzed the synteny between *IbSUSs*, *ItfSUSs* and *ItbSUSs*. The results indicated that most of *ItfSUSs* and *ItbSUSs* had one orthologous gene of *IbSUSs* apart from *ItfSUS3* on Chr03 with two orthologous genes (*IbSUS3* and *IbSUS4* on LG7) and *ItfSUS6* on Chr11 with two orthologous genes (*IbSUS5* on LG8 and *IbSUS6* on LG13). *IbSUS3* and *IbSUS4* were likely derived from a tandem duplication event and *IbSUS5* and *IbSUS6* had traces of segmental duplication (Figure 1d). These results showed that segmental and tandem duplications occurred in the process of evolution from diploid to hexaploid.

### 2.2. Phylogenetic Relationship Analysis of SUSs in Sweet Potato and Its Two Diploid Wild Relatives

In order to study the evolutionary relationships of SUSs in *I. batatas*, *I. trifida*, *I. triloba*, *Arabidopsis thaliana*, *Oryza sativa* and *Solanum tuberosum*, we constructed a phylogenetic tree for 43 SUSs of these six species (i.e., 9 in *I. batatas*, 7 in *I. trifida*, 7 in *I. triloba*, 6 in *A. thaliana*, 7 in *O. sativa* and 7 in *S. tuberosum*). According to the evolutionary distance, all the SUSs were divided into three groups and unevenly distributed on each branch of the phylogenetic tree (Figure 2). The specific distributions of SUSs were as follows (total: *I. batatas*, *I. trifida*, *I. triloba*, *A.thaliana*, *O. sativa,* and *S. tuberosum*): Group I (7:1, 1, 1, 2, 1, 1), Group II (17:4, 3, 3, 2, 3, 2), and Group III (19:4, 3, 3, 2, 3, 4) (Figure 2; Appendix A). These results indicated that all IbSUSs were clustered with their corresponding orthologs in *I. triloba* and *I. trifida*.

### 2.3. Conserved Motif and Exon–Intron Structure Analysis of SUSs in Sweet Potato and Its Two Diploid Wild Relatives

We analyzed sequence motifs in *IbSUSs*, *ItfSUSs* and *ItbSUSs* using the MEME website, and identified the ten conserved motifs (Figure 3a and Appendix A). Most of the *SUSs* contained these ten conserved motifs except for a few *SUSs* which were differentiated in the number and species of motifs only in *I. batatas*, such as *IbSUS3* (containing nine motifs except for 3), *IbSUS5* (containing eight motifs except for 2 and 9) and *IbSUS6* (containing nine motifs except for 2) (Figure 3a).

The exon-intron distributions of *IbSUS, ItbSUS* and *ItfSUS* were analyzed to better understand the evolution of the *IbSUS* gene family. In general, the number of exons and introns in *SUSs* showed small variations (Figure 3b). The number of exons ranged from 12 (*IbSUS7*, *IbSUS8*, *ItbSUS4*, *ItfSUS5*, *ItbSUS5*, *ItfSUS6* and *ItbSUS6*) to 16 (*IbSUS5*). In detail, some homologous *SUSs* had the same number of exons such as *IbSUS2*, *ItfSUS2* and *ItbSUS2* (all containing 15 exons) in Group I; *IbSUS3*, *ItfSUS3* and *ItbSUS3* (all containing 15 exons) in Group II; *IbSUS7*, *ItfSUS6* and *ItbSUS6* (all containing 12 exons) in Group III. However, some of them had more exons in *I. batatas* than in *I. trifida* and *I. triloba* such as *IbSUS1* (containing 15 exons), *ItfSUS1* (containing 14 exons) and *ItbSUS1* (containing 14 exons) in Group II; *IbSUS9* (containing 14 exons), *ItfSUS7* (containing 13 exons) and *ItbSUS7* (containing 13 exons); *IbSUS5* (containing 16 exons), *IbSUS6* (containing 13 exons), *ItfSUS4* (containing 12 exons) and *ItbSUS4* (containing 12 exons) in Group III (Figure 3b). These results indicated that the *SUS* gene family might become more complex in the evolution process.

### 2.4. Cis-Element Analysis in the Promoters of SUSs in Sweet Potato and Its Two Diploid Wild Relatives

The promoter *cis*-elements are tightly related to the gene function. In order to explore the regulatory mechanisms of *SUSs*, we analyzed the *cis*-elements in promoters of *IbSUSs, ItbSUSs* and *ItfSUSs* using their 2000 bp promoter regions. According to the functions of *cis*-elements, we divided them into six categories: core/binding, development, light, hormone, abiotic/biotic and other unknown elements (Figure 4). All of the *IbSUS* promoters contained a large number of core/binding elements including TATA-box and CAAT-box. The *IbSUS2* and *IbSUS3* promoters had also many AT-TATA-boxes. All of the *IbSUS* promoters possessed at least one development element. For example, the O_2_-site (zein metabolism regulatory element) was found in the promoters of *IbSUS3*, *IbSUS5* and *IbSUS9*; the CAT-box (associated with meristem formation and cell division) was found in the promoters of *IbSUS1* and *IbSUS8*; the GCN4 motif (control seed-specific expression) was found in the promoters of *IbSUS2* and *IbSUS7 *(Figure 4). The light-responsive elements TCT-motif, Box 4, G-box, AE-box, and AAGAA-motif existed in most of the *IbSUSs* promoters. The *IbSUSs* promoters contained abundant hormone-responsive elements, including ABA-responsive elements ABRE, ABRE4 and ABRE3a, JA-responsive elements CGTCA-motif and TGACG-motif, and SA-responsive elements TCA and TATC-box (Figure 4). Most of the *IbSUSs* promoters contained abiotic/biotic elements such as drought-responsive elements MYB and MYC, salt-responsive elements LTR and MBS, low temperature-responsive elements LTR, WRE3 and WUN motif, antioxidant response elements ARE and STRE (Figure 4).

There also were a large number of TATA-box and CAAT-box in all of the *ItfSUSs* and *ItbSUSs* promoters. Except for *ItfSUS3* and *ItbSUS3*, the promoters of other *ItfSUSs* and *ItbSUSs* contained at least one development element (Figure 4). Compared with the promoters of *IbSUSs*, the promoters of *ItfSUSs* and *ItbSUSs* contained less light-responsive elements, especially G-box and AE-box. The *ItfSUS1* and *ItbSUS1* promoters had no Box 4 and the *ItfSUS2* and *ItfSUS2* promoters had no AAGAA-motif, which existed in the promoters of homologous *IbSUSs*. In addition, Sp I and Box II existed only in the *IbSUS* promoters. The *ItfSUS* and *ItbSUS* promoters were also rich in hormone-responsive elements such as ABA-responsive element ABRE, JA-responsive elements CGTCA-motif and TGACG-motif, and SA-responsive element TCA (Figure 4). Interestingly, the promoters of *ItbSUS* contained more IAA-responsive element AuxRR-core than those of *IbSUSs* and *ItfSUSs*. The promoters of most of *ItfSUSs* and *ItbSUSs* had many abiotic/biotic elements, such as drought-responsive elements MYB and MYC, salt-responsive elements LTR and MBS, low temperature-responsive elements LTR, WRE3 and WUN motif, antioxidant response elements ARE and STRE (Figure 4). Taken together, these findings suggest that *SUSs* are involved in the regulation of plant growth and development, hormone crosstalk, and abiotic/biotic stress adaption in sweet potato and its two diploid wild relatives and *IbSUSs* may play more important roles in the development and light response of sweet potato.

### 2.5. Protein Interaction Network of IbSUSs in Sweet Potato

To investigate the potential regulatory network of IbSUSs, we constructed an IbSUSs interaction network based on *Arabidopsis* orthologous proteins (Figure 5a). Obviously, IbSUSs could not interact with each other (Figure 5a). IbSUSs could interact with glucose-1-phosphate adenylyltransferase (ADG1, APS2, APL1, APL2, APL3, and APL4), granule-bound starch synthase (GBSS1), phosphoglucomutase (PGMP and PGM2), sucrose phosphate synthase (SPS1F), UTP-glucose-1-phosphate uridylyl transferase (UGD1 and UGD2), cell wall invertase (CWINV4), 1,4-alpha-glucan-branching enzyme (SBE3) and glycosyl transferase (AT3G29320). IbSUS1, IbSUS4, IbSUS5 and IbSUS6 could interact with phosphoglucomutase (PGM3). IbSUS1 and IbSUS6 could interact with starch synthase (At1g11720). IbSUS2 and IbSUS3 could interact with UDP-sugar pyrophosphorylase (USP). IbSUS1, IbSUS2 and IbSUS3 could interact with UDP-glucose-hexose-1-phosphate uridylyl transferase (AT5G18200). These results have demonstrated that IbSUSs play important roles in the carbon metabolism of sweet potatoes.

In order to further explore the functions of IbSUSs, we conducted an Ontology (GO) pathway enrichment analysis for proteins involved in regulatory networks and classified 26 proteins into five significantly enriched GO pathways. Interestingly, the ‘starch biosynthetic process’ and ‘sucrose metabolic process’ were the dominant terms, which further proved the roles of IbSUSs in carbon metabolism (Figure 5b). ‘Response to abiotic stimulus’ was also involved in GO enrichment. Our data have proved that IbSUSs take on important responsibilities for starch biosynthesis and abiotic stress adaptation in sweet potatoes.

### 2.6. Expression Analysis of SUSs in Sweet Potato and Its Two Diploid Wild Relatives

#### 2.6.1. Expression Analysis in Various Tissues

The RNA-seq data of leaf, stem, fibrous root, and storage root tissues of sweet potato obtained at our laboratory were analyzed to investigate the potential biological functions of *IbSUSs*. We found that expression patterns of these *IbSUS*s were different in four tissues. *IbSUS2* belonging to Group I and *IbSUS5*, *IbSUS6*, *IbSUS7*, and *IbSUS9* belonging to Group III exhibited the highest expression level in storage roots; *IbSUS3*, *IbSUS4*, and *IbSUS8* belonging to Group II showed the highest expression level in stems, but the highest expression level of *IbSUS1* belonging to Group II was found in leaves (Figure 6a).

To study the expression patterns of *ItfSUSs* and *ItbSUSs*, we used RNA-seq data of six tissues (i.e., flower bud, flower, leaf, stem, root1, and root2) [33]. In *I. trifida*, *ItfSUS1*, *ItfSUS2*, *ItfSUS3* and *ItfSUS4* were highly expressed in flower buds; *ItfSUS5* and *ItfSUS6* were highly expressed in flowers; *ItfSUS1*, *ItfSUS3*, *ItfSUS4* and *ItfSUS7* were highly expressed in stems; *ItfSUS2* and *ItfSUS5* were highly expressed in root1 (Figure 6b). All the *ItfSUSs* had the low expression in leaves and root2. In *I. triloba*, *ItbSUS1*, *ItbSUS4* and *ItbSUS7* were highly expressed in stems; *ItbSUS2*, *ItbSUS5* and *ItbSUS6* were highly expressed in flower buds and flowers; *ItbSUS2* and *ItbSUS5* were highly expressed in roots (Figure 6c). All the *ItbSUSs* had low expression in leaves and no expression of *ItbSUS3* was tested. Above all, these results have shown that *SUSs* possess different expression patterns and play important roles in the growth and development of sweet potato and its two diploid wild relatives.

#### 2.6.2. Expression Analysis at Different Developmental Stages of Storage Roots in Sweet Potato

In order to understand the roles of *SUSs* in storage root development of sweet potato, we further performed qRT-PCR to evaluate the expression levels of *IbSUSs* at different developmental stages of storage roots (i.e., 20, 30, 40, 50, 60, 70, 80, 90, 100 and 130 DAP: day after planting) of sweet potato line H283 with high starch content which was selected from F_1_ individuals from a cross between Xushu 18 and Xu 781 [35]. As shown in Figure 7, *IbSUS2*, *IbSUS4*, *IbSUS8* and *IbSUS9* were highly expressed at 50 DAP (rapid thickening and initial starch accumulation stage); *IbSUS4*, *IbSUS7*, *IbSUS8* and *IbSUS9* were highly expressed during 80 to 90/100 DAP (rapid starch accumulation); *IbSUS1*, *IbSUS3* and *IbSUS5* exhibited a sharp expression level at 130 DAP (harvest stage with the large production of soluble sugars). In addition, *IbSUS6* showed a low expression level during the entire growth and development of sweet potato. These results indicated that different *IbSUSs* might play important roles in the development and starch and sugar accumulation at different developmental stages of sweet potato storage roots.

#### 2.6.3. Expression Analysis in Sweet Potato Lines with Different Starch Contents

As described above, *IbSUSs* might take on important responsibilities for starch biosynthesis in sweet potatoes (Figure 5). To further investigate the roles of *IbSUSs* in starch biosynthesis of sweet potato, we analyzed the expression levels of *IbSUSs* in H283 and H471 with high starch content, M638 and M88 with medium starch content, and L23 and L408 with low starch content, which were selected from F_1_ individuals from a cross between Xushu 18 and Xu 781 [35]. As a whole, *IbSUSs* were highly expressed in the high starch lines, in particular, *IbSUS1*, *IbSUS2*, *IbSUS5, IbSUS6*, and *IbSUS7* exhibited significantly higher expression levels in the high starch lines than in the medium/low starch lines. *IbSUS3*, *IbSUS4*, *IbSUS8* and *IbSUS9* had no obvious trend in expression (Figure 8).

#### 2.6.4. Expression Analysis under Drought and Salt Stresses

To investigate the potential roles of *IbSUSs* in drought and salt stress responses, the expression patterns of *IbSUSs* were analyzed using the RNA-seq data of a drought-tolerant line Xu55-2 under PEG6000 stress and the RNA-seq data of a salt-sensitive variety Lizixiang and a salt-tolerant line ND98 under NaCl stress [36,37]. Under PEG6000 stress, *IbSUS2*, *IbSUS5*, *IbSUS6* and *IbSUS7* were upregulated and others were downregulated or did not show significant changes in Xu55-2 (Figure 9). Under NaCl stress, *IbSUS3*, *IbSUS4* and *IbSUS8* were downregulated in ND98 and upregulated in Lizixiang and others were upregulated in both ND98 and Lizixiang (Figure 9).

Next, the expression patterns of *ItfSUSs* and *ItbSUSs* were analyzed using the RNA-seq data of *I. trifida* and *I. triloba* under drought and salt stresses [33]. In *I. trifida*, *ItfSUS1* was upregulated only under NaCl stress, *ItfSUS2*, *ItfSUS5* and *ItfSUS6* were upregulated under mannitol and NaCl stresses and others were downregulated or did not show significant changes under mannitol and NaCl stresses (Appendix A). In *I. triloba*, *ItbSUS2* and *ItbSUS5* were upregulated under mannitol and NaCl stresses, *ItbSUS6* was upregulated only under NaCl stress, and others were downregulated or did not show significant changes under mannitol and NaCl stresses (Appendix A). Taken together, these results indicated that some *SUSs* were involved in response to drought and salt stresses in sweet potato and its two diploid wild relatives.

#### 2.6.5. Expression Analysis in Response to Hormones

To analyze the potential biological functions of *IbSUSs* in the hormone crosstalk of sweet potato, we conducted qRT-PCR to evaluate the expression levels of *IbSUSs* in sweet potato line H283 in response to hormones, including ABA, GA3, IAA, MeJA, and SA (Figure 10). Under ABA treatment, *IbSUS2* and *IbSUS8* were upregulated and others were downregulated or did not show significant changes (Figure 10a). Under GA3 treatment, *IbSUS2*, *IbSUS3*, *IbSUS4*, *IbSUS5*, *IbSUS6*, *IbSUS8* and *IbSUS9* were upregulated and *IbSUS1* and *IbSUS7* were downregulated (Figure 10b). Under IAA treatment, *IbSUS1*, *IbSUS2*, *IbSUS4*, *IbSUS5* and *IbSUS8* were upregulated and others were downregulated or did not exhibit significant changes (Figure 10c). Under MeJA treatment, all *IbSUSs* were upregulated and their expression levels reached the peak at 6 h except *IbSUS5* (Figure 10d). When treated with SA, all *IbSUSs* were upregulated and *IbSUS4* peaked at1 h and others peaked at 24 h (Figure 10e).

The expression patterns of *ItfSUSs* and *ItbSUSs* were also analyzed using the RNA-seq data of *I. trifida* and *I. triloba* under ABA, GA3 and IAA treatments [33]. As shown in Appendix A, under ABA treatment, *ItfSUS7* were downregulated and others were upregulated. Under GA3 treatment, *ItfSUS1*, *ItfSUS3*, *ItfSUS4* and *ItfSUS7* were upregulated and others were downregulated or did not show significant changes. All *ItfSUSs* were downregulated when treated with IAA. In *I. triloba*, *ItbSUS1*, *ItbSUS2*, *ItbSUS4*, *ItbSUS5* and *ItbSUS6* were upregulated, *ItbSUS7* were downregulated, and *ItbSUS3* showed no expression under ABA treatment. *ItbSUS4*, *ItbSUS6* and *ItbSUS7* were upregulated and others were downregulated or did not show significant changes by GA3. When treated with IAA, only *ItbSUS2* was upregulated and others were downregulated (Appendix A). Therefore, these results indicated that different *SUSs* exhibited different expression profiles in response to hormones and they might participate in the crosstalk between various hormones in sweet potato and its two diploid relatives.

## 3. Discussion

### 3.1. Evolution of SUSs in Sweet Potato and Its Two Diploid Wild Relatives

In recent years, the *SUS* gene family has been identified in various plant species through comparative genome approaches. The genomes of model species *Arabidopsis* [11] and rice [13] both contain six *SUSs*. The number of *SUSs* is eight in diploid cotton (*G. arboretum* and *G. raimondii*) and 15 in tetraploid cotton (*G. hirsutum*), respectively, and it is thought that *SUSs* of *G. hirsutum* derived from those of its diploid ancestors and one *SUS* group underwent expansion during cotton evolution [14]. In our study, nine, seven, and seven *SUSs* were identified in *I. batatas*, *I. trifida*, and *I. triloba*, respectively. The number of *SUSs* identified in sweet potato was two more than that in its two diploid wild relatives. According to the chromosomal localization and phylogenetic relationships of *SUSs*, the homologs among *I. batatas*, *I. trifida* and *I. triloba* were located on similar sites (Figure 1 and Figure 2), suggesting that *SUSs* of sweet potato derived from those of its diploid ancestors. Two additional *SUSs* located on LG7 and LG13 in sweet potato were homologous to *SUS3* and *SUS5* in *I. trifida* and *I. triloba*, respectively. Based on the syntenic analysis (Figure 1d), we speculate that segmental and tandem duplications could lead to the increased copy number of *SUSs* in sweet potatoes, which might bring about functional redundancy and sub/neo-functionalization [38].

Furthermore, ten conserved motifs were identified in all of the SUSs except for IbSUS3, IbSUS5, and IbSUS6 (Figure 3a). IbSUS3 was homologous to IbSUS4, ItfSUS3, and ItbSUS3, but it did not have motif3. IbSUS5 and IbSUS6 were homologous to ItfSUS4 and ItbSUS4, but IbSUS5 lacked motif2 and motif9 and IbSUS6 lacked motif2. These results indicated that IbSUS3, IbSUS5 and IbSUS6 went through structure changes in the process of evolution, which might result in their functional differences.

Gene exon/intron structures are typically conserved among homologous genes of a gene family [39]. In our study, the majority of homologous *SUSs* had the same number of exons and introns in *I. batatas*, *I. trifida* and *I. triloba*, but some of them were different in exon-intron structures, for example, *IbSUS1* (15 exons), *ItfSUS1* (14 exons) and *ItbSUS1* (14 exons) in Group II; *IbSUS9* (14 exons), *ItfSUS7* (13 exons) and *ItbSUS7* (13 exons) in Group III; *IbSUS5* (16 exons), *IbSUS6* (13 exons), *ItfSUS4* (12 exons) and *ItbSUS4* (12 exons) in Group III (Figure 3b). Therefore, some *IbSUSs* have more exons than *ItfSUSs* and *ItbSUSs*, suggesting that exon-intron structures could lead to more functional diversification [40]. Taken together, these results reveal that *IbSUSs* are more complex in evolution and could participate in more biological processes compared with *ItfSUSs* and *ItbSUSs*.

### 3.2. IbSUSs Are Involved in Storage Root Development and Starch Biosynthesis in Sweet Potato

As the marker for sink strength in plants, SUSs play important roles in sink organ development and their activities are generally much higher in developing seeds and fruits than in leaves [1]. In maize, *ZmSUS1* mutant (*sh1*) had less starch content and weight in seeds, but the transactivation of *Sus1* and *Sus2* by endosperm-specific transcription factor Opaque2 (O2) could help the endosperm filling [11,20]. In potatoes, suppressed expression of *StSUS4* decreased the tuber dry weight and starch content and enhanced StSUS4 activity increased the tuber yield and starch content [41,42]. Inhibition of the activity of SiSUS3 in tomatoes and CsSUS4 in cucumbers reduced the number, size and weight of flowers and fruits [43,44]. Overexpression of *GhSUS*2 in cotton and *BeSUS5* in poplar enhanced cellulose production, cell wall thickness and fiber quality [25,27]. In addition, by controlling the channeling of incoming sucrose into starch and cell wall biosynthesis, different members of a given SUS family may play different roles in nonphotosynthetic cells. For instance, *ZmSUS1* participated in starch biosynthesis in maize seed, *ZmSUS2* contributed to endosperm cell wall biosynthesis, and *ZmSUS3* may take part in the construction of basal endosperm transfer cells [11,22].

In this study, we discovered that IbSUSs could interact with many proteins related to storage root development and starch biosynthesis (Figure 5a). GO enrichment analysis showed that ‘starch biosynthetic process’ and ‘sucrose metabolic process’ were the dominant terms of these related proteins (Figure 5b). Because of this, IbSUSs may participate in storage root development and starch biosynthesis of sweet potatoes. Interestingly, *SUSs* in *I. trifida* and *I. triloba* were highly expressed in flower buds, flowers, root1 or stems (Figure 6). *IbSUS2*, *IbSUS5*, *IbSUS6*, *IbSUS7* and *IbSUS9* were highly expressed in storage roots, suggesting that they may be involved in starch biosynthesis of storage roots; *IbSUS3*, *IbSUS4* and *IbSUS8* were highly expressed in stems and may participate in transport and assimilation of carbohydrates (Figure 6).

As the most important harvest organ of sweet potato, storage roots are the main sink that accepts most carbohydrates. At different developmental stages of storage roots, *IbSUSs* exhibited different expression levels. *IbSUS2*, *IbSUS4*, *IbSUS8* and *IbSUS9* were highly expressed at 50 DAP; *IbSUS4*, *IbSUS7*, *IbSUS8* and *IbSUS9* were highly expressed during 80 to 90/100 DAP; *IbSUS1*, *IbSUS3*, and *IbSUS5* were sharply expressed at 130 DAP (Figure 7). Meanwhile, *IbSUS1*, *IbSUS2*, *IbSUS5, IbSUS6* and *IbSUS7* exhibited much higher expression in the high starch lines than in the medium/low starch lines (Figure 8). Therefore, *IbSUS2*, *IbSUS5* and *IbSUS7* may play more important roles in storage root development and starch biosynthesis, which have the potential to be used for increasing yield and starch content in sweet potato.

### 3.3. SUSs Regulate Response to Drought and Salt Stresses in Sweet Potato and Its Two Diploid Wild Relatives

*SUSs* are reported to regulate plant response to drought stress. In *Arabidopsis*, drought treatment induced the expression of *AtSS1* [11]. Two *SUSs* (*HvSS1* and *HvSS3*) in barley and one *SUS* (*HbSS5*) in rubber trees significantly responded to drought stress [45,46]. However, there is no research indicating that *SUSs* regulate response to salt stress. In this study, ‘Response to abiotic stimulus’ was involved in GO enrichment (Figure 5b). Furthermore, *IbSUS2*, *IbSUS5*, *IbSUS6* and *IbSUS7* in sweet potato, *ItfSUS2*, *ItfSUS5* and *ItfSUS6* in *I*. *trifida*, and *ItbSUS2* and *ItbSUS5* in *I. triloba* were upregulated under drought stress (Figure 9 and Appendix A). Therefore, the *SUSs*, especially *SUS2*, may play crucial roles in response to drought stress in sweet potato and its two diploid relatives.

Under NaCl stress, *IbSUS3*, *IbSUS4* and *IbSUS8* were downregulated in ND98 but upregulated in Lizixiang and others were upregulated in both ND98 and Lizixiang (Figure 9). *ItfSUS1*, *ItfSUS2*, *ItfSUS5* and *ItfSUS6* in *I. trifida* and *ItbSUS2*, *ItbSUS5* and *ItbSUS6* in *I. triloba* were upregulated under NaCl stress (Appendix A). These results suggest that *SUS2, SUS5* and *SUS6* may regulate the response to salt stress in sweet potato and its two diploid relatives, but their salt tolerance function might have weakened during the evolution of sweet potato.

### 3.4. SUSs Participate in Hormone Crosstalk in Sweet Potato and Its Two Diploid Wild Relatives

In tomatoes, suppression of *SlSUS1*, *SlSUS3* and *SlSUS4* altered auxin signaling, indicating their possible roles in the regulation of plant growth and leaf morphology [47]. However, there is no report on *SUSs* regulating other hormones. In this research, we found abundant ABA-, JA-, and SA-responsive elements in the *IbSUS* promoters (Figure 4). The *ItfSUSs* and *ItbSUSs* promoters were also rich in these hormone-responsive elements (Figure 4). Furthermore, most of the *IbSUSs* were induced by ABA, GA3, IAA, MeJA and SA (Figure 10). Most of the *ItfSUSs* and *ItbSUSs* were also induced by ABA, GA3 and IAA (Appendix A). Especially, *IbSUS2* and its homologous *ItfSUS2* and *ItbSUS2* were upregulated under ABA treatment; *IbSUS5* and *IbSUS6* and its homologous *ItfSUS4* and *ItbSUS4* and *IbSUS9* and its homologous *ItfSUS7* and *ItbSUS7* were upregulated when treated with GA3. These results suggest that the *SUSs* may participate in the hormone crosstalk in sweet potato and its diploid wild relatives.

## 4. Materials and Methods

### 4.1. Identification of SUSs

All protein sequences of *I. batatas*, *I. trifida* and *I. triloba* were downloaded from the *Ipomoea* Genome Hub (https://ipomoea-genome.org/, accessed on 2 March 2023) and the Sweet potato Genomics Resource (http://sweetpotato.plantbiology.msu.edu/, accessed on 2 March 2023). The Hidden Markov Model (HMM) profiles of the core sucrose synthase domain (PF00862) and Glycosyltransferases domain (PF00534) from the Pfam database (http://pfam.xfam.org/, accessed on 2 March 2023) were downloaded and used to survey all SUS proteins in *I. batatas*, *I. trifida* and *I. triloba*. All the presumed SUSs were checked using SMART (http://smart.embl-heidelberg.de/, accessed on 2 March 2023) and Conserved Domains Database (CDD, https://www.ncbi.nlm.nih.gov/Structure/cdd/wrpsb.cgi, accessed on 2 March 2023).

### 4.2. Chromosomal Distribution of SUSs

*IbSUSs*, *ItfSUSs* and *ItbSUSs* were separately mapped to the *I. batatas*, *I. trifida* and *I. triloba* chromosomes based on the chromosomal locations provided in the *Ipomoea* Genome Hub (https://ipomoea-genome.org/, accessed on 2 March 2023) and Sweet potato Genomics Resource (http://sweetpotato.plantbiology.msu.edu/, accessed on 2 March 2023). The visualization was generated using the TBtools software v1.120 (South China Agricultural University, Guangzhou, China) [48].

### 4.3. Property Prediction of SUSs

The MW, theoretical *p*I, unstable index and hydrophilic of the SUSs were calculated using ExPASy (https://www.expasy.org/, accessed on 3 March 2023). The subcellular localization was predicted using WoLF PSORT (https://www.genscript.com/wolf-psort.html, accessed on 3 March 2023).

### 4.4. Phylogenetic Analysis of SUSs

The SUS sequences of *A. thaliana*, *I. batatas*, *I. triloba*, *I. trifida*, *O. sativa* and *S. tuberosum* were downloaded from the National Center for Biotechnology Information (https://www.ncbi.nlm.nih.gov/, accessed on 3 March 2023) and aligned with ClustalX. The aligned sequences were imported into MEGA11 to create a phylogenetic tree using the neighbor-joining method with 1000 bootstrap replicates [49]. The phylogenetic tree was constructed by iTOL (http://itol.embl.de/, accessed on 13 March 2023).

### 4.5. Domain Identification and Conserved Motif Analysis of SUSs

The conserved motifs of *SUSs* were analyzed using MEME software (https://meme-suite.org/meme/, accessed on 4 March 2023) and the maximum number of motif parameters was set to 10 [50]. The conserved domain structures of *SUSs* were visualized using the TBtools software v1.120 (South China Agricultural University, Guangzhou, China).

### 4.6. Exon–Intron Structure and Promoter Analyses of SUSs

The exon–intron structures of *SUSs* were obtained and visualized by the TBtools software v1.120 (South China Agricultural University, Guangzhou, China). The *cis*-elements in the approximately 2000 bp promoter regions of *SUSs* were predicted by PlantCARE (http://bioinformatics.psb.ugent.be/webtools/plantcare/html/, accessed on 4 March 2023) [51].

### 4.7. Protein Interaction Network and GO Enrichment Analysis of SUSs and Regulated Proteins

The protein interaction network of SUSs was built using STRING (https://cn.string-db.org/, accessed on 5 March 2023) based on *Arabidopsis* homologous proteins. The network map was constructed using Cytoscape 3.2 [52]. GO enrichment analysis of SUSs and regulated proteins was implemented using the cluster Profiler R package [53].

### 4.8. Transcriptome Analysis

The RNA-seq data used for expression analysis of *SUSs* in *I. batatas* were obtained at our laboratory [36,37] and those of *SUSs* in *I. trifida* and *I. triloba* were downloaded from the Sweet Potato Genomics Resource (http://sweetpotato.plantbiology.msu.edu/, accessed on 4 March 2023). The expression levels of *SUSs* were calculated as fragments per kilobase of exon per million fragments mapped (FPKM) and the heat maps of expression were constructed by TBtools software v1.120 (South China Agricultural University, Guangzhou, China).

### 4.9. qRT-PCR Analysis of SUSs

The storage roots of H283 sampled at the different development stages (20, 30, 40, 50, 60, 70, 80, 90, 100 and 130 DAP) and H283, H471, M638, M88, L23 and L408 sampled at 90 DAP were used for analyzing the expression of *IbSUS*. The leaves of the one-month-old in vitro-grown H283 plants were treated with 100 µM ABA, 100 µM GA3, 100 µM IAA, 100 µM MeJA and 100 µM SA, and then sampled at 0, 1, 3, 6, 12 and 24 h after treatment for expression analysis of *IbSUSs* in response to different hormones. The total RNA was extracted with the TRIzol method (Invitrogen, Carlsbad, CA, USA). The qRT-PCR was performed on a 7500 Real-Time PCR system (Applied Biosystems, Foster City, CA, USA) using the SYBR detection protocol (TaKaRa, Kyoto, Japan). Sweet potato *actin* gene (GenBank AY905538) was used as an internal control. The specific primers used for the qRT-PCR analysis were listed in Appendix A. The values were determined from three biological replicates consisting of pools of three plants and analyzed using the comparative C_T_ method [54]. The heat maps of the gene expression were constructed using the TBtools software v1.120 (South China Agricultural University, Guangzhou, China). The significant difference of each *IbSUS* was analyzed at *p* < 0.05 based on one-way ANOVA followed by posthoc Tukey’s test.

## 5. Conclusions

Nine, seven and seven *SUSs* were identified in the cultivated sweet potato and its two diploid wild relatives *I. trifida* and *I. triloba*, respectively. Their protein physicochemical properties, chromosomal localization, phylogenetic relationship, gene structure, promoter *cis*-elements, protein interaction network and expression patterns were systematically analyzed. Segmental and tandem duplications could lead to the increased copy number of *SUSs* in sweet potato, which might bring about functional redundancy and sub/neo-functionalization. *IbSUSs* have more exons than *ItfSUSs* and *ItbSUSs* which could participate in more biological processes compared with *ItfSUSs* and *ItbSUSs*. The *SUSs* were highly expressed in sink organs. *IbSUSs* might play vital roles in storage root development and starch biosynthesis in sweet potatoes and *IbSUS2*, *IbSUS5* and *IbSUS7* as candidate genes that can affect storage root development and starch biosynthesis are worth further research (Figure 11). Besides, the *SUSs* could also respond to drought and salt stress responses. However, the salt tolerance function of *SUSs* might have weakened during the evolution of sweet potatoes (Figure 11). They also took part in hormone crosstalk especially ABA and GA (Figure 11). This work provides valuable insights into the structure and function of *SUSs* and candidate genes for improving yield, starch content and abiotic stress tolerance in sweet potato.

## Figures and Tables

**Figure 1 ijms-24-12493-f001:**
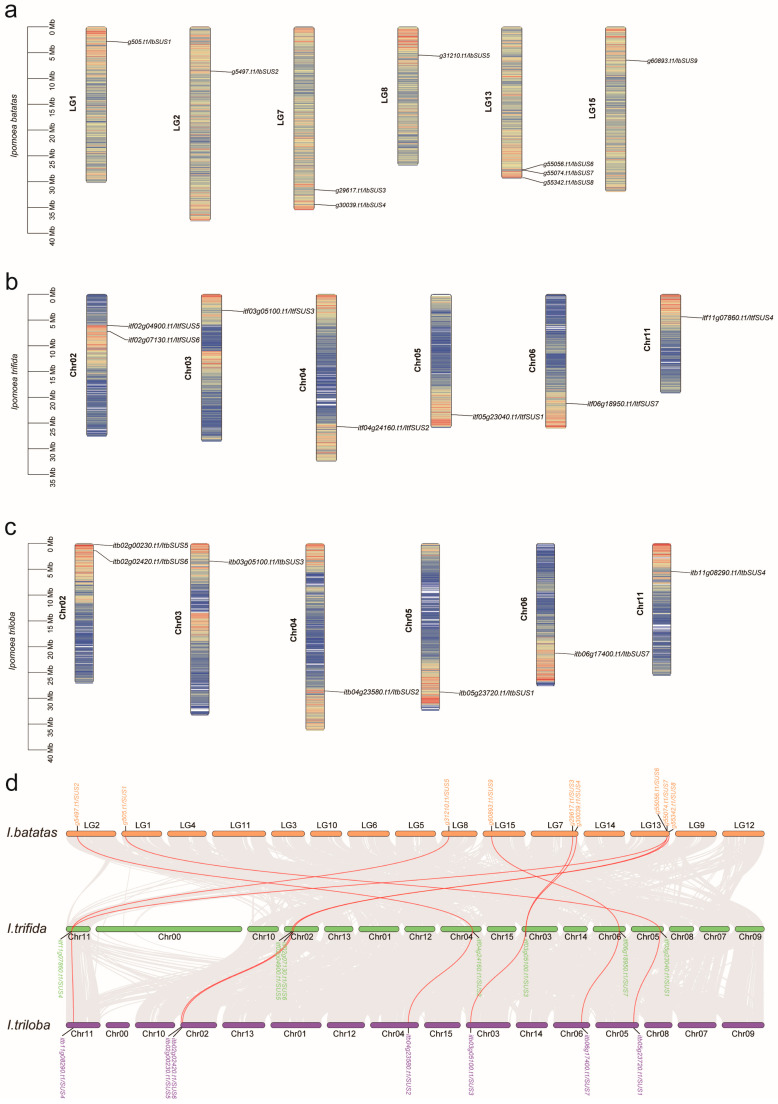
Chromosomal localization and distribution of *SUSs* in *I. batatas* (**a**), *I. trifida* (**b**), and *I. triloba* (**c**). The bars represent chromosomes. The chromosome numbers are displayed on the left side, and the gene names are displayed on the right side. The relative chromosomal localization of each *SUS* gene is marked on the black line of the right side and indicated by the unit Mbp. (**d**) Syntenic analysis of *I. batatas*, *I. trifida* and *I. triloba SUS*s. Chromosomes of *I. batatas*, *I. trifida* and *I. triloba* are shown in different colors. The approximate positions of *IbSUSs*, *ItfSUSs* and *ItbSUSs* are marked with short black lines on the chromosomes. Red curves denote the syntenic relationships between *I. batatas* and *I. trifida SUSs*.

**Figure 2 ijms-24-12493-f002:**
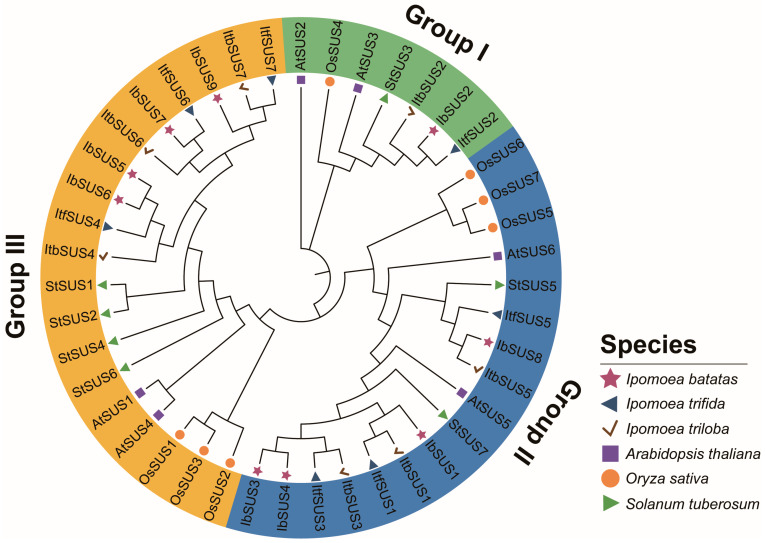
Phylogenetic analysis of SUSs in *I. batatas*, *I. triloba*, *I. trifida*, *A. thaliana*, *O. sativa* and *S. tuberosum*. Based on the evolutionary distance, a total of 43 SUSs were divided into three groups (groups I, II, and III, filled with green, blue, and orange, respectively). The claret stars represent nine IbSUSs in *I. batatas*. The dark blue triangles represent seven ItfSUSs in *I. trifida*. The brown check marks represent seven ItbSUSs in *I. triloba*. The purple squares represent six AtSUSs in *A. thaliana*. The orange circles represent six OsSUSs in *O. sativa*. The green triangles represent seven StSUSs in *S. tuberosum*.

**Figure 3 ijms-24-12493-f003:**
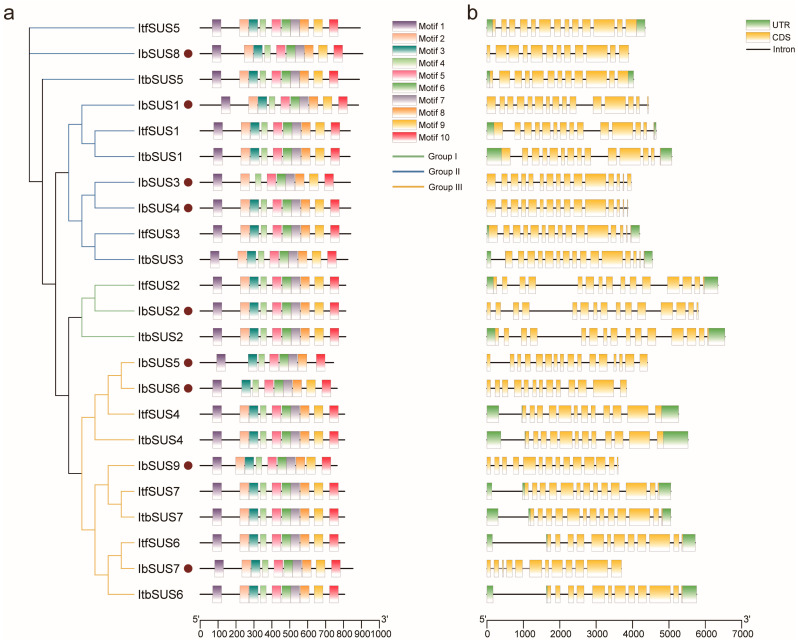
Conserved motifs and exon–intron structure analysis of *SUSs* in *I. batatas*, *I. trifida*, and *I. triloba*. (**a**) The phylogenetic tree showed that SUSs were divided into three subgroups and the ten conserved motifs were shown in different colors. The brown circles represent IbSUSs. (**b**) Exon–intron structures of *SUSs*. The green boxes, yellow boxes, and black lines represent UTRs, exons and introns, respectively.

**Figure 4 ijms-24-12493-f004:**
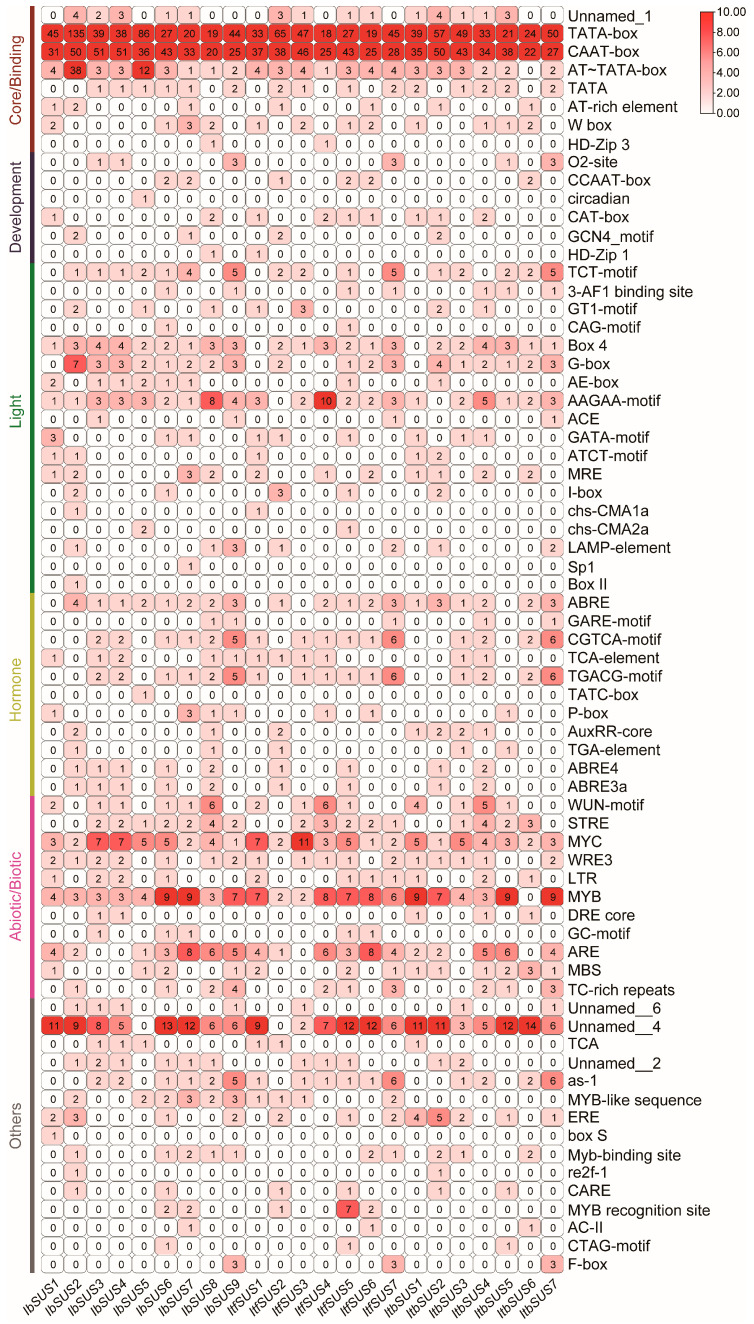
*Cis*-element analysis in the promoters of *SUSs* from *I. batatas*, *I. trifida* and *I. tri loba*. The *cis*-elements were divided into six broad categories. The degree of red colors represents the number of *cis*-elements in the promoters of *SUSs*.

**Figure 5 ijms-24-12493-f005:**
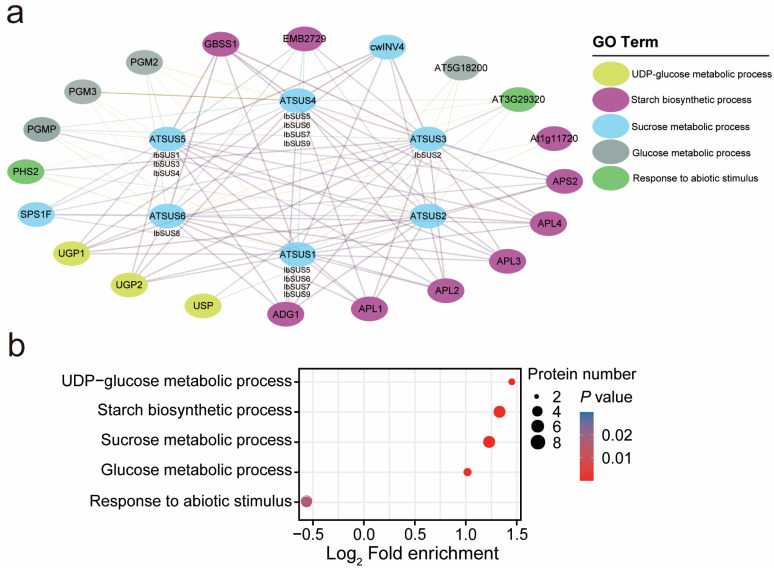
Functional interaction networks of IbSUSs in *I. batatas* according to orthologues in *Arabidopsis* and GO enrichment analysis of proteins in networks. (**a**) Network nodes represent proteins, and lines represent protein–protein associations. The thickness and color of lines represent the interaction strength. (**b**) GO enrichment analysis of proteins included in networks. Fisher’s exact test is used to perform GO pathway enrichment analysis. Significantly enriched KEGG pathways (*p* < 0.05) are shown.

**Figure 6 ijms-24-12493-f006:**
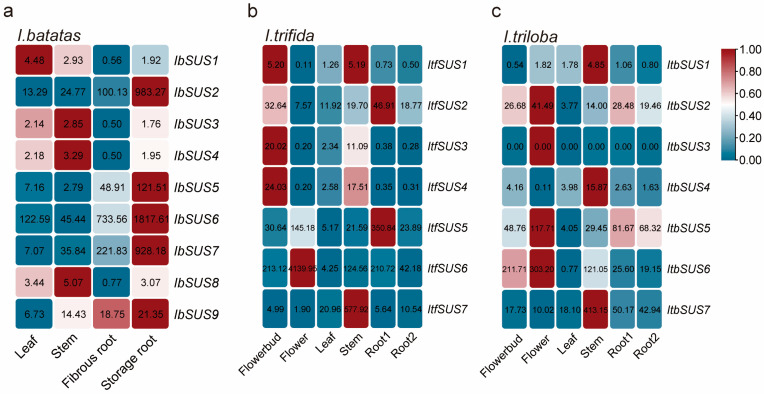
Expression analysis of *SUSs* in different tissues of *I. batatas*, *I. trifida*, and *I. triloba* using RNA-seq. (**a**) Expression patterns of *IbSUSs* in the leaf, stem, fibrous root and storage root of *I. batatas*. (**b**) Expression patterns of *ItfSUSs* in the flower bud, flower, leaf, stem, root1 and root2 of *I. trifida*. (**c**) Expression patterns of *ItbSUSs* in the flower bud, flower, leaf, stem, root1 and root2 of *I. triloba*. The FPKM values are shown in the boxes.

**Figure 7 ijms-24-12493-f007:**
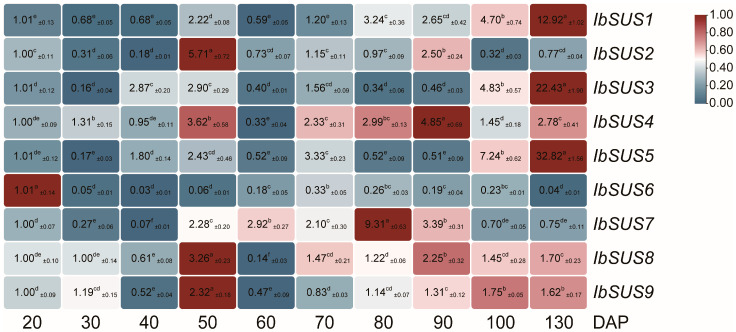
Expression analysis of *IbSUSs* at different developmental stages of the H283 storage roots (i.e., 20, 30, 40, 50, 60, 70, 80, 90, 100 and 130 DAP) using qRT-PCR. The values were determined by qRT-PCR from three biological replicates consisting of pools of three plants and the results were analyzed using the comparative C_T_ method. The expression level at 20 DAP is determined as “1”. Fold change ± SD is shown in the boxes. Different lowercase letters indicate a significant difference of each *IbSUS* at *p* < 0.05 based on one-way ANOVA followed by posthoc Tukey’s test.

**Figure 8 ijms-24-12493-f008:**
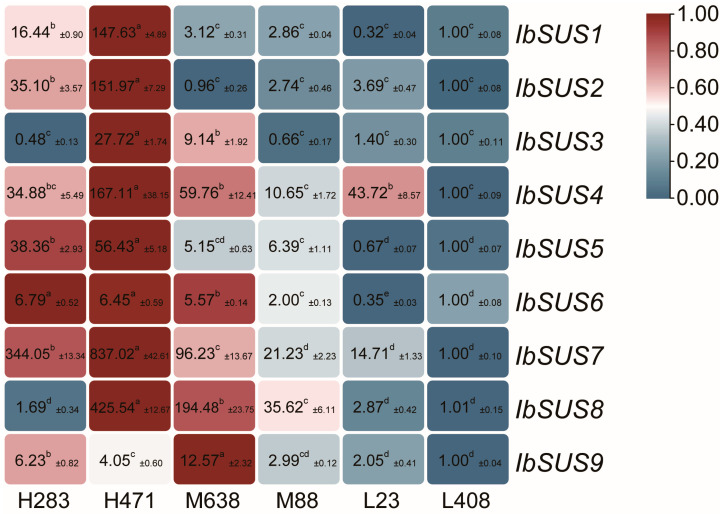
Expression analysis of *IbSUSs* in sweet potato lines with different starch contents. The values were determined by qRT-PCR from three biological replicates consisting of pools of three plants and the results were analyzed using the comparative C_T_ method. The expression level of L408 is considered as “1”. Fold change ± SD is shown in the boxes. Different lowercase letters indicate a significant difference of each *IbSUS* at *p* < 0.05 based on one-way ANOVA followed by post-hoc Tukey’s test.

**Figure 9 ijms-24-12493-f009:**
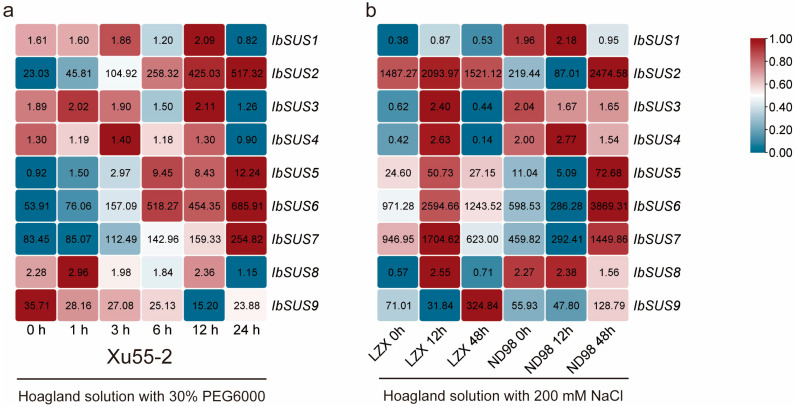
Expression analysis of *IbSUSs* in sweet potato under drought and salt stresses as determined by RNA-seq. (**a**) Expression analysis of *IbSUSs* in the drought-tolerant line Xu55-2 under PEG6000 stress. (**b**) Expression analysis of *IbSUSs* in the salt-sensitive variety Lizixiang (LZX) and salt-tolerant line ND98 under NaCl stress. The FPKM values are shown in the boxes.

**Figure 10 ijms-24-12493-f010:**
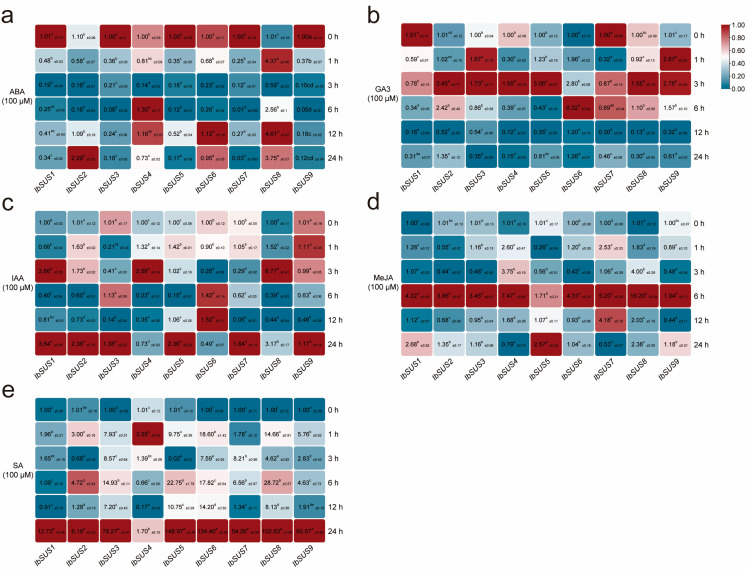
Expression analysis of *IbSUSs* in response to different hormones in sweet potato line H283. (**a**) ABA. (**b**) GA3. (**c**) IAA. (**d**) MeJA. (**e**) SA. The values were determined by qRT-PCR from three biological replicates consisting of pools of three plants and the results were analyzed using the comparative C_T_ method. The expression at 0 h in each treatment is determined as “1”. Fold change ± SD is shown in the boxes. Different lowercase letters indicate a significant difference of each *IbSUS* at *p* < 0.05 based on one-way ANOVA followed by posthoc Tukey’s test.

**Figure 11 ijms-24-12493-f011:**
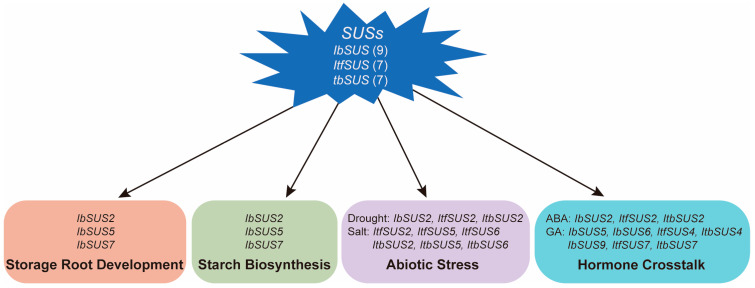
A conclusive graph of functions of *SUSs* in sweet potato and its two diploid wild relatives *I. trifida* and *I. triloba*.

**Table 1 ijms-24-12493-t001:** Characteristics of *SUSs* in *I. batatas*, *I. trifida* and *I. triloba*.

Gene ID	Gene Name	Genomic Length (bp)	CDS (bp)	Protein Size (aa)	MW (kDa)	*p*I	Instability	GRAVY	Subcellular Location
g505	*IbSUS1*	4689	2649	882	99.99	6.21	40.26	−0.312	cytoplasm
g5497	*IbSUS2*	6116	2436	811	92.13	6.02	38.17	−0.275	chloroplast
g29617	*IbSUS3*	4187	2514	837	94.75	6.14	36.59	−0.351	cytoplasm
g30039	*IbSUS4*	4091	2520	839	94.95	6.33	38.73	−0.33	cytoplasm
g31210	*IbSUS5*	4954	2232	743	84.78	6.11	36.16	−0.202	plastid
g55056	*IbSUS6*	4794	2292	763	87.82	6.17	33.53	−0.298	mitochondria
g55074	*IbSUS7*	5591	2556	851	97.49	6.16	38.91	−0.256	mitochondria
g55342	*IbSUS8*	4125	2721	906	101.58	6.99	39.82	−0.214	chloroplast
g60893	*IbSUS9*	4931	2292	763	87.7	6.18	35.82	−0.261	cytoplasm
itf05g23040	*ItfSUS1*	4648	2511	836	94.86	6.02	38.89	−0.326	cytoplasm
itf04g24160	*ItfSUS2*	6346	2436	811	92.21	5.99	37.62	−0.281	chloroplast
itf03g05100	*ItfSUS3*	4194	2520	839	94.88	6.22	39.2	−0.336	cytoplasm
itf11g07860	*ItfSUS4*	5265	2418	805	92.60	5.93	34.11	−0.279	cytoplasm
itf02g04900	*ItfSUS5*	4342	2679	892	99.89	6.40	38.81	−0.248	chloroplast
itf02g07130	*ItfSUS6*	5723	2418	805	92.49	5.95	35.03	−0.25	cytoplasm
itf06g18950	*ItfSUS7*	5052	2418	805	92.71	5.96	33.65	−0.27	cytoplasm
itb05g23720	*ItbSUS1*	5077	2508	835	94.88	6.10	38.94	−0.332	cytoplasm
itb04g23580	*ItbSUS2*	6536	2436	811	92.09	6.02	38	−0.277	chloroplast
itb03g05100	*ItbSUS3*	4547	2472	823	93.27	6.30	38.87	−0.325	chloroplast
itb11g08290	*ItbSUS4*	5524	2418	805	92.58	6.02	33.8	−0.271	cytoplasm
itb02g00230	*ItbSUS5*	4028	2664	887	99.29	6.51	37.07	−0.236	chloroplast
itb02g02420	*ItbSUS6*	5760	2418	805	92.46	5.93	35.3	−0.241	cytoplasm
itb06g17400	*ItbSUS7*	5052	2418	805	92.74	5.96	33.76	−0.267	cytoplasm

CDS, coding sequence; MW, molecular weight; *p*I, isoelectric point; GRAVY, grand average of hydropathicity.

## Data Availability

The data presented in this study are available on request from the corresponding author.

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
