# Peer review of "Genome-Wide Identification and Expression Analysis of the Sucrose Synthase Gene Family in Sweet Potato and Its Two Diploid Relatives"

_ijms, 2023, doi:10.3390/ijms241512493_

Round 1
Reviewer 1 Report
In connection with the fact that SUS family genes (SUSs) have been identified in several plants, but they have not been explored in sweet potato, in this research, several SUSs were identified in the cultivated sweet potato. This studies provides new insights for further understanding the functions of SUSs and candidate genes for improving yield, starch content, and also abiotic stress tolerance in sweet potato. In the experiments SUSs were identified from sweet potato (I. batatas), I.trifida, and I. triloba, and then their protein physicochemical properties, chromosomal localization, phylogenetic relationships, gene structure, promoter cis-elements and protein interaction network were analyzed. The methods used are correct. The results were documented properly. Statistical tests were chosen correctly. Noteworthy is the very good graphical representation of the results. The English language requires minor corrections by a native speaker.
The English language requires minor corrections by a native speaker.
Author Response
Response to Reviewer 1 Comments
Reviewer 1:
In connection with the fact that SUS family genes (SUSs) have been identified in several plants, but they have not been explored in sweet potato, in this research, several SUSs were identified in the cultivated sweet potato. This study provides new insights for further understanding the functions of SUSs and candidate genes for improving yield, starch content, and also abiotic stress tolerance in sweet potato. In the experiments SUSs were identified from sweet potato (I. batatas), I.trifida, and I. triloba, and then their protein physicochemical properties, chromosomal localization, phylogenetic relationships, gene structure, promoter cis-elements and protein interaction network were analyzed. The methods used are correct. The results were documented properly. Statistical tests were chosen correctly. Noteworthy is the very good graphical representation of the results. The English language requires minor corrections by a native speaker.
Response:
We thank the reviewer for this comment. Based on the comment, we have made careful modifications in the discussion as follow:
“and seven SUSs were identified in the cultivated sweet potato (Ipomoea batatas, 2n = 6x = 90) as well as its two diploid wild relatives I. trifida (2n = 2x = 30) and I. triloba (2n = 2x = 30),” (Lines 16-17)
“The results indicated that the SUS gene family underwent segmental and tandem duplications during its evolution.” (Lines 21-22)
“As a result, sucrose biosynthesis is essential for the growth and development of plants [2,6,7].” (Lines 35-36)
“According to Zhang et al. [34], nine SUSs existed in the genome of sweet potato and the expression level of IbSus6 was higher in storage roots of Kokei 14 than in those of its mutant.” (Lines 74-76)
“responses were carried out using qRT-PCR or RNA-seq.” (Line 82)
“In order to study the evolutionary relationships of SUSs in I. batatas, I. trifida, I. triloba,” (Line 144)
“According to the evolutionary distance, all the SUSs were divided into three groups and unevenly distributed on each branch of the phylogenetic tree (Figure 2).” (Lines 147-149)
“We analyzed sequence motifs in IbSUSs, ItfSUSs and ItbSUSs using the MEME website, and identified the ten conserved motifs (Figure 3a and Figure S1).” (Lines 165-166)
“IbSUS3, IbSUS4, IbSUS8 and IbSUS9 had no obvious trend in expression (Figure 8).” (Line 327)

Reviewer 2 Report
1. I'm not sure if IJMS still has an interest to publish this kind of research using bioinformatics and qRT-PCR for a "preliminary" exploration of candidate genes. This type of research can not provide "in-depth" insights into molecular mechanisms of anything unless the authors do some actual test for the gene functions.
2. The manuscript title point out the main theme is to compare different species of sweet potato, but I did not read too much content for this. What is the difference of SUSs between them? What is the contribution of this kind of results?
3. L17: "n" and "x" should be in italics throughout the manuscript.
4. L22-23: The authors should describe these results according to statistics and point out which effects have significant differences.
5. Keywords: These should be critical terms but did not show in the manuscript title.
6. Conclusion: Too hollow and should be more specific and not look so similar to the abstract. The authors should point out the main progress of this study on the molecular mechanisms of SUSs.
7. A conclusive graph for the entire story particularly for presenting the mechanisms will be helpful.
Author Response
Response to Reviewer 2 Comments
Reviewer 2:
- I'm not sure if IJMS still has an interest to publish this kind of research using bioinformatics and qRT-PCR for a "preliminary" exploration of candidate genes. This type of research cannot provide "in-depth" insights into molecular mechanisms of anything unless the authors do some actual test for the gene functions.
Thank you for your suggestion. Our work is the first systematic analysis of SUS genes in sweet potato and its two diploid relatives. Our results elucidate their protein physicochemical properties, chromosomal localization, phylogenetic relationships, gene structure, promoter cis-elements and protein interaction network and found that during evolution, segmental and tandem duplications occurred in the process of evolution from diploid to hexaploid resulting in the difference of distribution and physicochemical properties between sweet potato and its two diploid relatives (Lines: 18-20, 129-130). Then we used qRT-PCR and transcriptome data to explore the function of SUSs and found that they play a role in storage root development, starch biosynthesis, drought and salt stress responses, and hormone crosstalk (Lines: 22-24, 576-577). These results provide a clearer direction for our future in-depth research on gene function and have strong guiding significance. Moreover, our research on the molecular mechanisms of SUS genes is also ongoing.
In recent years, many articles with similar ideas and methods have also been published on IJMS (International Journal of Molecular Sciences), such as JAZ (ijms-22189786), CDPK (ijms-23063088) and SWEET (ijms-232415848) family genes in sweet potato; DUF506 gene family in Arabidopsis (ijms-222111442); FKBP gene family in wheat (ijms-232314501); 14-3-3 gene family in mango (ijms-23031593); F-box family in poplar (ijms-231810934); GATA gene family in buckwheat (ijms-232012434); HD-Zip transcription factor family in apple (ijms-23052632) and NF-Y transcription factor family in alfalfa (ijms-23126426). Therefore, we believe that our research is in line with the interests of IJMS.
- The manuscript title point out the main theme is to compare different species of sweet potato, but I did not read too much content for this. What is the difference of SUSs between them? What is the contribution of this kind of results?
We apologize for the inconvenience. As the genetic background of cultivated sweet potato (Ipomoea batatas (L.) Lam., 2n = B1B1B2B2B2B2 = 6x = 90) is complex. Its genome is difficult to assemble and is highly polymorphic. In fact, the plant morphology of cultivated hexaploid sweet potato differs greatly from that of its diploid relatives, especially its diploid relatives cannot form tuberous roots. However, the functional roles of most gene families are still poorly understood in sweet potato.
In our study, we for the first time systematically identified, investigated and compared the chromosome localization, phylogenetic relationship, gene structure, and expression pattern of 23 SUSs (i.e., 9 in I. batatas, 7 in I. trifida, and 7 in I. triloba) from the cultivated hexaploid sweet potato and its two diploid relatives. The results suggested that (1) the number and distribution of SUSs of sweet potato were different with those of its two diploid relatives (Figure 1; Lines 89-90, 119-128, 399-401); (2) homologous SUSs with more exon and introns in hexaploid sweet potato are evolutionally more complex than its two diploid relatives (Figure 3; Lines 176-182, 421-423); (3) cis-elements analysis showed that the promoters of ItfSUSs and ItbSUSs contained less development elements and light-responsive elements than IbSUSs. IbSUSs may play more important roles in the development and light response of sweet potato. (Figure 4; Lines 213-216, 229-230); (4) compared to sweet potato, its diploid relatives cannot form storage roots. Because of this, SUSs in sweet potato especially IbSUS2, IbSUS5 and IbSUS7 may play more important roles in storage root development and starch biosynthesis, which have the potential to be used for increasing yield and starch content in sweet potato. (Figure 7-8; Lines 294-332, 442-461). All these results indicated that there was differentiation between homologous SUSs, and each SUS gene played different vital roles in growth and development, starch biosynthesis and hormone crosstalk between sweet potato and its two diploid relatives.
With respect, we hope that this study will serve as a reference for sweet potato researchers, and call for more studies on gene families in hexaploid sweet potato in the future.
- L17: "n" and "x" should be in italics throughout the manuscript.
Thank you very much for your suggestions. Based on the comments, we have made careful modifications in the revised manuscript.
Please refer to the red section on lines 16, 17 and 64.
- L22-23: The authors should describe these results according to statistics and point out which effects have significant differences.
We thank the reviewer for this suggestion. We use min-max normalization to process the data and then our results were tested by one-way ANOVA followed by post-hoc Tukey’s test (p<0.05) to identify significant difference. These results are showed by heatmap with different color. Data with fold change and lowercase letters indicated a significant difference also shown in the boxes. Because of this, our results were documented properly and statistical tests were chosen correctly. We provide a detailed description of the results in the Results and Discussion section. According to our results, we can draw a conclusion that the SUSs were highly expressed in sink organs and might play vital roles in storage root development, starch biosynthesis, drought and salt stress responses, and hormone crosstalk.
Compared to I. trifida and I. triloba, sweet potato can form storage roots which are the most important harvest organ of sweet potato (452-453) and starch is the main component of storage roots (67-68). Because of this, roles in storage root development and starch biosynthesis of SUSs in sweet potato will become the mainly focus of future attention and IbSUS2, IbSUS5 and IbSUS7 as candidate genes that can affect storage root development and starch biosynthesis are worth for further research. As suggested, we add the information as follows:
“The SUSs were highly expressed in sink organs. The IbSUSs especially IbSUS2, IbSUS5 and IbSUS7 might play vital roles in storage root development and starch biosynthesis. The SUSs could also response to drought and salt stress responses, and took part in hormone crosstalk.”
(Please see lines 22-24.)
- Keywords: These should be critical terms but did not show in the manuscript title.
Thank you very much for your suggestion. Firstly, in the manuscript title, we directly include two keywords: “sweet potato” and “sucrose synthase”. Next, two diploid relatives in the title includes two keywords: I. trifida and I. triloba, and expression analysis includes four aspects: storage root development, starch biosynthesis, abiotic stress and hormone crosstalk. Finally, this type of articles often used this type of titles such as “Genome-Wide Identification and Expression Analysis of SWEET Family Genes in Sweet Potato and Its Two Diploid Relatives” (ijms-232415848), “Genome-Wide Identification and Expression Analysis of the 4-Coumarate: CoA Ligase Gene Family in Solanum tuberosum” (ijms-24021642) and “Genome-Wide Identification, Evolution, and Expression Analyses of AP2/ERF Family Transcription Factors in Erianthus fulvus” (ijms-24087102). Therefore, the manuscript title showcases these keywords.
- Conclusion: Too hollow and should be more specific and not look so similar to the abstract. The authors should point out the main progress of this study on the molecular mechanisms of SUSs.
We thank the reviewer for this great suggestion. Based on this suggestion, we have provided a more detailed description of the manuscript as follow, especially in terms of molecular mechanisms:
“Nine, seven and seven SUSs were identified in the cultivated sweet potato and its two diploid wild relatives I. trifida and I. triloba, respectively. Their protein physicochemical properties, chromosomal localization, phylogenetic relationship, gene structure, promoter cis-elements, protein interaction network and expression patterns were systematically analyzed. Segmental and tandem duplications could lead to the increased copy number of SUSs in sweet potato, which might bring about functional redundancy and sub/neo-functionalization. IbSUSs have more exons than ItfSUSs and ItbSUSs which could participate in more biological processes compared with ItfSUSs and ItbSUSs. The SUSs were highly expressed in sink organs. IbSUSs might play vital roles in storage root development and starch biosynthesis in sweet potato and IbSUS2, IbSUS5 and IbSUS7 as candidate genes that can affect storage root development and starch biosynthesis are worth for further research (Figure 11). Besides, the SUSs could also response to drought and salt stress responses (Figure 11). But salt tolerance function of SUSs might have weakened during the evolution of sweet potato. They also took part in hormone crosstalk especially ABA and GA (Figure 11). This work provides valuable insights into the structure and function of SUSs and candidate genes for improving yield, starch content and abiotic stress tolerance in sweet potato.”
(Please see lines 573-589.)
- A conclusive graph for the entire story particularly for presenting the mechanisms will be helpful.
We thank the reviewer for this great suggestion. As suggested, we added a conclusive graph and figure legend in our revised manuscript as follow:
Figure 11. A conclusive graph of functions of SUSs in sweet potato and its two diploid wild relatives I. trifida and I. triloba.
(Please see lines 591-594.)

Reviewer 3 Report
The study “Genome-wide identification and expression analysis of the sucrose synthase gene family in sweet potato and its two diploid relatives” by Zhicheng Jiang et al is a high quality work done to a very good standard. All sections of the manuscript are properly designed according to the rules of the journal. The results are quite good, and for the first time the sucrose synthase gene family of an important agricultural crop has been thoroughly characterized, and it is this gene family that makes almost the main contribution to the filling of sweet potato tissues and organs with an important product, starch.
The disadvantage of this work may be the excessive importance, given to bioinformatics data, although, following the goals set by the authors, all of them has been achieved. In addition, the expression analysis based on qRT-PCR provides a fairly reliable confirmation of the expression of individual copies of the sucrose synthase genes in specific localizations. Moreover, the authors revealed the degree of expression of various genes in different localizations. The established connection between the expression of sucrose synthase genes and accompanying genes involved in the same process is also important. Of course, it would be good to look at the individual expression of the promoters of various sucrose synthase genes with marker genes, but this is a separate, very large work that other researchers can do based on the results obtained by the authors.
A small remark concerns the end of the Introduction section. Here, the authors describe their work as already done. But the results should have been moved to the end of the Discussion section or the Conclusion section. Here, the authors should have rewritten this paragraph and clearly indicated, for example: “The aim of the study was ...” or “The goal of the work was ...”
In general, this manuscript should be accepted for publication in the IJMS with minor revisions.
Author Response
Response to Reviewer 3 Comments
Reviewer 3:
The study “Genome-wide identification and expression analysis of the sucrose synthase gene family in sweet potato and its two diploid relatives” by Zhicheng Jiang et al is a high quality work done to a very good standard. All sections of the manuscript are properly designed according to the rules of the journal. The results are quite good, and for the first time the sucrose synthase gene family of an important agricultural crop has been thoroughly characterized, and it is this gene family that makes almost the main contribution to the filling of sweet potato tissues and organs with an important product, starch.
The disadvantage of this work may be the excessive importance, given to bioinformatics data, although, following the goals set by the authors, all of them has been achieved. In addition, the expression analysis based on qRT-PCR provides a fairly reliable confirmation of the expression of individual copies of the sucrose synthase genes in specific localizations. Moreover, the authors revealed the degree of expression of various genes in different localizations. The established connection between the expression of sucrose synthase genes and accompanying genes involved in the same process is also important. Of course, it would be good to look at the individual expression of the promoters of various sucrose synthase genes with marker genes, but this is a separate, very large work that other researchers can do based on the results obtained by the authors.
A small remark concerns the end of the Introduction section. Here, the authors describe their work as already done. But the results should have been moved to the end of the Discussion section or the Conclusion section. Here, the authors should have rewritten this paragraph and clearly indicated, for example: “The aim of the study was ...” or “The goal of the work was ...”
In general, this manuscript should be accepted for publication in the IJMS with minor revisions.
Response:
Thank you very much for your suggestions. Based on the comments, we have made careful modifications in the revised manuscript as follow.
“The aim of this study was to provide new insights for further understanding the functions of SUSs and candidate genes for improving yield, starch content, and abiotic stress tolerance in sweet potato.”
(Please see lines 82-84.)

Reviewer 4 Report
Dear Athors,
The manuscript deals with an interesting topic, offering interesting results. The research results are quite well documented. The manuscript is well-written and I do not have any comments.
Author Response
Response to Reviewer 4 Comments
Reviewer 4:
Dear Athors,
The manuscript deals with an interesting topic, offering interesting results. The research results are quite well documented. The manuscript is well-written and I do not have any comments.
Response:
Thank you very much for your affirmation of our research and article.

Round 2
Reviewer 2 Report
I don't have further questions.